# A Guideline for Contextual Adaptation of Community-Based Health Interventions

**DOI:** 10.3390/ijerph19105790

**Published:** 2022-05-10

**Authors:** Zinzi E. Pardoel, Sijmen A. Reijneveld, Maarten J. Postma, Robert Lensink, Jaap A. R. Koot, Khin Hnin Swe, Manh Van Nguyen, Eti Poncorini Pamungkasari, Lotte Tenkink, Johanna P. M. Vervoort, Johanna A. Landsman

**Affiliations:** 1Department of Health Sciences, University Medical Center Groningen, University of Groningen, Hanzeplein 1, 9700 RB Groningen, The Netherlands; s.a.reijneveld@umcg.nl (S.A.R.); m.j.postma@rug.nl (M.J.P.); j.a.r.koot@umcg.nl (J.A.R.K.); lottetenkink@gmail.com (L.T.); j.p.m.vervoort@umcg.nl (J.P.M.V.); j.a.landsman@umcg.nl (J.A.L.); 2Department of Pharmacology and Therapy, Faculty of Medicine, Universitas Airlangga, Surabaya 60115, Indonesia; 3Centre of Excellence in Higher Education for Pharmaceutical Care Innovation, Universitas Padjadjaran, Bandung 45363, Indonesia; 4Department of Economics, Econometrics & Finance, Faculty of Economics and Business, University of Groningen, 9747 AE Groningen, The Netherlands; b.w.lensink@rug.nl; 5HelpAge International, Yangon 11081, Myanmar; swe.khinhnin@helpagemyanmar.org; 6HelpAge International, Hanoi 1000, Vietnam; manhnv@helpagevn.org; 7Department of Public Health, Faculty of Medicine, Universitas Sebelas Maret, Surakarta 57126, Indonesia; etiponco@staff.uns.ac.id

**Keywords:** cultural context, guideline, adaptation, participatory action research, co-creation, Positive Health, community-based health interventions

## Abstract

In Southeast Asia, community-based health interventions (CBHIs) are often used to target non-communicable diseases (NCDs). CBHIs that are tailored to sociocultural aspects of health and well-being: local language, religion, customs, traditions, individual preferences, needs, values, and interests, may promote health more effectively than when no attention is paid to these aspects. In this study, we aimed to develop a guideline for the contextual adaption of CBHIs. We developed the guideline in two stages: first, a checklist for contextual and cultural adaptation; and second, a guideline for adaptation. We performed participatory action research, and used the ‘Appraisal of Guidelines for Research & Evaluation (AGREE) II’ tool as methodological basis to develop the guideline. We conducted a narrative literature review, using a conceptual framework based on the six dimensions of ‘Positive Health’ and its determining contexts to theoretically underpin a checklist. we pilot tested a draft version of the guideline and included a total of 29 stakeholders in five informal meetings, two stakeholder meetings, and an expert review meeting. This yielded a guideline, addressing three phases: the preparation phase, the assessment phase, and the adoption phase, with integrated checklists comprising 34 cultural and contextual aspects for the adaption of CBHIs based on general health directives or health models. The guideline provides insight into how CBHIs can be tailored to the health perspectives of community members, and into the context in which the intervention is implemented. This tool can help to effect behavioral change, and improve the prevention and management of NCDs.

## 1. Introduction

In Southeast Asia, community-based health interventions (CBHIs) are often implemented to target non-communicable diseases (NCDs) [1,2,3]. There is some indication that CBHIs—that are tailored to sociocultural aspects of health and well-being: local language, religion, customs, traditions, individual preferences, needs, values, and interests—may promote health more effectively than when no attention is paid to these aspects [4,5,6]. CBHIs are complex social processes involving multiple components such as screening, physical exercise and education, and are aimed at preventing illness and unhealthy behavior and promoting the well-being of various population groups [1]. These CBHIs address health risk behaviors that are major determinants of NCDs, such as tobacco smoking, alcohol, unhealthy diet, and physical inactivity. CBHIs aim to improve these behaviors by using primary health care to increase people’s knowledge about health, solidarity, self-reliance, social support, and synergy [7]. Health behaviors are determined to a significant degree by health perception, which is highly context-dependent [8], i.e., formed in the demographic, healthcare, cultural and social contexts to which an individual belongs [9,10]. For this reason, CBHIs must be culturally adapted to the contexts of the participants to effectively bring about behavioral change and improve the prevention and management of NCDs [11].

In 1948, the World Health Organization (WHO) defined health as “a state of complete physical, mental and social well-being and not merely the absence of disease or infirmity” [12]. Although this aspirational definition of health is accepted worldwide, it does not always align with health perceptions. Huber’s concept of ‘Positive Health’ [13,14] provides a theoretical framework to accommodate the impact of context on peoples’ individual health perception, making it a helpful guide for the adaptation of such interventions. The concept consists of six major dimensions: *bodily functions*, *mental well-being*, *meaningfulness*, *participation*, *daily functioning*, and *quality of life*. According to this concept, health is established in and influenced by peoples’ health perception, which is in turn associated with their cultural, social, healthcare, and demographical environments (see Figure 1). Therefore, incorporating the dimensions of ‘Positive Health’ can help to adapt CBHIs to the context in which they are to be applied, thus, enhancing the promotion, prevention, and management of health. 

CBHIs are commonly developed based on validated directives, models, and approaches, such as the WHO‘s Integrated Care for Older People (ICOPE)—community-level interventions for managing the decline in intrinsic capacity [16]. 

To date, several studies address the guidance for the adaptation of existing interventions to context. First, the ADAPT guideline [17] offers a framework to help researchers, policy and practice stakeholders, funders, and journal editors in undertaking and assessing the adaptation of interventions for a new context, and reporting these transparently. Second, the report of Graig and colleagues provides guidance on how context should be taken into account, from priority setting and intervention development to the design and conduct of evaluations and reporting, synthesis, and knowledge exchange [18]. These two guidelines provide extensive and in-depth information for the adaptation and reporting process of (health) interventions. However, to our knowledge, there are no concrete, practical, scientifically based guidelines available to adapt CBHIs to a specific context, even though this is considered to be an important aspect of most guidelines [19]. This study, therefore, aimed to develop such a concrete and practical guideline, and was part of the EU-H2020 funded project, “Scaling up Non-communicable disease interventions in South East Asia” (SUNI SEA), taking place in Myanmar, Indonesia, and Vietnam, and coordinated from The Netherlands.

## 2. Materials and Methods

### 2.1. Design

We developed the guideline in two stages with six steps: first, a checklist for contextual and cultural adaptation; and second, a guideline for adaptation, which includes the checklist (see Figure 2). The conceptual framework of ‘Positive Health’ and its determining contexts formed the theoretical basis for our narrative literature review and for the inclusion of contextual and cultural aspects in the guideline. We derived the steps per stage using ‘Appraisal of Guidelines for Research & Evaluation (AGREE) II’ [20], and step-by-step assessed the issues raised by this instrument. The purpose of AGREE-II is to provide a framework to assess and achieve quality, and a methodological strategy for the development of guidelines. We then used a participatory action research (PAR) [21] approach to shape co-creation [22]: collaborative knowledge generation by academics working jointly with other stakeholders. 

### 2.2. Sample and Procedure

We collected data from multiple sources, using a number of samples (see Table 1). In both stages, 29 stakeholders were involved; data were collected during two-hour meetings, held online via platforms WebEx and Microsoft Teams, and audio recorded. All data were transcribed verbatim and analyzed using qualitative content analysis of transcripts. 

The first stage, development of the checklist, involved two steps. In step 1, we conducted a narrative literature review, based on the concept ‘Positive Health’ and its determining contexts. The second step involved informal meetings. For the latter, we used a PAR approach [21], which has two distinguishing characteristics: participation of stakeholders as partners in the research process, and commitment to action for social change [23]. In every step of our action research, we reflected on previous steps and findings, using McIntyre’s ‘Recursive Process of PAR’ (2008) [21]. This iterative process includes six activities: questioning, reflecting, investigating, developing, implementing and refining. 

The second stage of the study involved four steps, resulting in the final guideline. In every step we followed the criteria of the AGREE-II [20], which can be applied in any disease area, targeting any step in the healthcare continuum, including health promotion, screening, or interventions. Because the AGREE-II was developed specifically for clinical guidelines, we adjusted the tool to make it more compatible for our research: we converted conditions or health issues into aspects of context. We then used the tool to guide the development of the guideline, covering all of its domains with several strategies. The AGREE-II consists of 23 key items within six domains. We covered the first domain, scope and purpose, by adding to the guideline a description of the objectives, the questions, and the target group. For the second domain, stakeholder involvement, we held meetings with various stakeholders. For the third domain, rigor of development, we conducted mixed-method research; for the fourth, clarity of presentation, we consulted with various stakeholders and experts to assure transparency, readability, and clarity. For the fifth domain, applicability, we conducted pilot testing and multiple evaluations of the guideline; this resulted in an extensive description of applicability, as well as possible barriers and facilitators. Finally, for the sixth domain, editorial independence, we recorded and addressed the issues of funding body and competing interests.

In the last step, we synthesized, reviewed and reflected upon all data using the conceptual framework. This led to the finalized guideline with the checklist.

**Table 1 ijerph-19-05790-t001:** Overview of data samples and characteristics.

Data Sample	Narrative Literature Review	Informal Meetings	Stakeholder Meetings	Expert Meeting
Characteristics				
Phase	Phase 1 development of checklists	Phase 1 development of checklists	Phase 2 development of the guideline. After pilot testing the guideline.	Phase 2 after stakeholder meetings
Number of articles/persons	13	9	15	5
Sampling method	Databases: PubMed, Google, Psych info and snowball method	Members of SUNISEA consortium	Convenience sampling [24], starting with stakeholders involved in the pilot trainings	Convenience sampling the expert pool of HelpAge International
Involvement with CHBIs	-	Development, research and/or implementation of CBHIs	Observers in pilot trainings or involved in the research or implementation of interventions	Country directors of NGOs involved in CBHIs
Gender	-	3 males/6 females	5 males/10 females	2 males/3 females
Years/Age range	1993–2020	28–63 years	2 –53 years	32–55 years
Countries	Asian countries	2 Indonesia, 1 Myanmar, 2 Vietnam, 4 Netherlands	10 Indonesia 5 Vietnam	1 Moldova, 1 The Philippines, 1 Sri Lanka, 1 Cambodia, 1 Vietnam
Period	April–May 2020	May–June 2020	January–February 2021	September 2021

The two stages of our study included the following activities:

### 2.3. Stage 1: Development of a Checklist for Contextual Adaptation of the CBHIs

In the first stage, we developed a checklist. In the first step, we conducted a narrative literature review [25] in April and May to establish focus for the checklist. In this review, articles were included that covered contextual and cultural aspects of Asia within the six dimensions of ‘Positive Health’. Moreover, only the literature in English was included. Articles were excluded when they did not cover Asian countries nor cultural and contextual aspects within the six domains. We used the databases PubMed, Google, and Psych info, combining different key words such as “Positive Health”, “Health Perception”, “Cultural aspects”, “Context”, and “Interventions”. To find the literature, we used the snowball method, using the bibliography or footnotes of a paper to identify additional papers [26]. Based on the findings of this review, we developed a first draft of the checklist.

The second step involved consultations via informal meetings with stakeholders (n = 9) who were working on the development, research, and/or implementation of the CBHIs. The draft of the checklist was disseminated in the SUNI-SEA consortium in May and June 2021. Based on the feedback in the meetings, we began the development of the guideline. 

### 2.4. Stage 2: Development of a Guideline for Application of the Checklist

In the second stage, we developed a guideline using steps 3, 4, 5, and 6. In the third step, we pilot tested the draft version of the guideline, including the checklist, during the development of materials, and implementation of a community-based training (CBT) for organizing and giving the CBHIs. In December 2020 and January 2021 in Indonesia and Vietnam, during the implementation of the CBT, independent observers used the checklist and guideline to examine the cultural and contextual aspects of the training. In Vietnam, the draft version was pilot tested during two training sessions, with 35 members and one observer per session. In Indonesia, the draft version was pilot tested once within a group of 20 members and one observer. Observers filled in evaluation forms regarding their experiences with the implementation of the checklist and guideline. In the fourth step, in January and February 2021, we organized stakeholder meetings. The topic list for the stakeholder meeting was based on the five areas of focus for feasibility studies by Bowen and colleagues [27], namely: (1) practicality (e.g., “To what extent can guideline be carried out with intended participants using existing means, resources, and circumstances?”); (2) adaption (e.g., “To what extent does the guideline perform when changes are made for a new format or with a different population?”); (3) acceptability (e.g., “To what extent is the guideline judged as suitable, satisfying, or attractive to program deliverers or program recipients?”); (4) implementation (e.g., “To what extent can the guideline be successfully delivered to intended participants in some defined, but not fully controlled, context?”); and (5) demand (e.g., To what extent is the guideline likely to be used). In the fifth step, in September 2021, we held an expert review meeting. In the expert meeting, the topics were based on the five focus areas regarding feasibility studies as proposed by Bowen and on the results of the stakeholder meetings. The findings of the stakeholder and expert meetings were categorized under facilitators and barriers based on the revisions made to the guideline and checklist. As a sixth step, we synthesized, reviewed, and reflected upon all data within the conceptual framework. This step concluded the co-creation phase, and the researchers finalized the guideline with the checklist. 

## 3. Results

### 3.1. Stage 1: Development of the Checklist

The findings of the first step, the literature review, are presented in Appendix B, Table A1, illustrating the dimensions related to cultural or contextual aspects. We found 13 articles, including reviews, explorative studies and qualitative studies, covering Asian contextual and cultural aspects within the dimensions of ‘Positive Health’. The perception of ‘bodily functions’ in Southeast Asia is comprehensive; physical health is conceptualized as the harmony and unity of mind, body, and soul [28]. As Ravindran et al. (2012) explain, “upset in body balance is the common way to look at disease; it refers to the belief that a healthy body is in a state of balance. When the body gets out of balance, illness results” [28]. A number of factors can disturb this balance, such as certain foods, medications, herbs, or strong emotions. Therefore, what you eat or emotionally feel can directly influence your organs and your bodily functioning [29]. A contextual aspect within the ‘mental well-being’ dimension is the stigmatization of mental illness. In Southeast Asia, emotional expression is commonly considered to be personal weakness, and can contribute to being stigmatized with mental illness [30]. Such stigma is also grounded in the rigidity of restraint societies in Southeast Asia, where the predominantly practiced Buddhism looks upon mental illness as suffering caused by one’s own past misdeeds [31]. For the ‘meaningfulness’ dimension, we found that in different cultural societies, factors that give life meaning are often found in spiritual and religious beliefs [32]. Most countries in Southeast Asia are multicultural, with many minority groups, resulting in a variety of religions [30]; the most commonly practiced religion is Buddhism, whose basic principles are often familiar to people of other religions. The ‘participation’ dimension depends on a balance between opportunities and limitations [14]. Associated with this balance is the ability to participate and being actively involved in ordinary family and community activities [33]. Southeast Asian countries often have a high intergenerational co-residence, where children take care of their parents [30]. The Southeast Asian elderly participate mainly by giving advice to family and community members. Their accumulated life wisdom and spiritual capacities make this advice highly appreciated [33]. Further, contextual aspects for ‘daily functioning’ include current or past work-related activities and availability of/barriers to healthy food. A study by Nilsson et al. (2005) indicated that in Southeast Asia, being functional in daily life is ‘having the strength and physical ability to work’ [33]. Another determiner of daily functioning is having good health, which in turn can be related to food [33]. According to Huber, one aspect of the ‘quality of life’ dimension is happiness. Uschida et al. (2004) indicated that the cultural meaning of happiness in Southeast Asia is defined mainly in terms of interpersonal connectedness [34].

In the second step, we collected feedback from several stakeholders (n = 9) in informal meetings. This yielded additional new aspects of context: adult friendly methods, cultural and individual exercise options, and the role of a trainer/implementer (e.g., role model for a healthy lifestyle). However, as some topics turned out to be ambiguous and difficult to understand, we added an appendix with definitions and meanings to the guideline. A further important finding from the meetings was the need for a guideline with practical information on how to apply the checklist. We, therefore, jointly agreed to include instructions for the use of the checklist in our guideline. 

In the second stage, we developed a guideline.

### 3.2. Stage 2: Development of the Guideline

The draft of the guideline included basic practical information on how to apply the checklist. In the third step, we pilot tested the guideline. The guideline, including the checklist, was perceived as a helpful tool for addressing important contextual aspects during the development of training materials, and for monitoring the implementation of the training. In both countries, materials and trainings were adjusted based on the checklist. In Indonesia, information about traditional medicines and herbs (e.g., traditional therapy and alternative medicines) and the stigmatization of illnesses and mental health was added to the training. In Vietnam, the guideline was pilot tested in different areas (i.e., rural and urban), resulting in a different adaptation of the training to an urban or to a rural context, even though it was implemented in the same country. 

In the fourth step, we held stakeholder meetings to discuss the results of pilot testing the guideline (n = 15). We categorized the results under perceived facilitators of and barriers to the guideline and made appropriate revisions. Appendix C, Table A2, gives an overview of these results, including quotations that illustrate the findings. The main facilitator mentioned by the stakeholders was that the checklist and guideline were useful for adapting a (medical) message to the appropriate culture and context. Moreover, the stakeholders indicated that the guideline should be made adaptable to different contexts (e.g., countries, areas, groups), because several contextual aspects vary within different contexts. Based on this finding, we added an explanation—that cultural and contextual aspects differ per context, that some aspects are more important than others, and that some aspects can be irrelevant for certain contexts. The main barriers mentioned by the stakeholders were time allocation and translation, which we discussed further in the expert review meeting.


*“Some medical words were used in the materials that health practitioners use. However, not all community members know these words. Based on the checklist, we found this and revised this.”.*
(Stakeholder from Indonesia)

In the fifth step, several international experts (n = 5) discussed the guideline with the checklist, providing an in-depth review of the guideline. Their views and ideas converged with those in the stakeholder meetings, as shown in Appendix C. The experts found the guideline to be innovative and essential for developing the CBHIs. One of their main recommendations was to involve stakeholders: community members, representatives, government and funding parties. Involvement of stakeholders can reduce the barriers of translation and time. However, effective involvement of stakeholders requires appropriate allocation of budget, and also time. A new barrier mentioned by the experts was implementers not always being role models for the participants of the CBHIs and they recommended to carefully reconsider the word ‘role model’. 


*“It is too sensitive that an implementer or a trainer should be a role model for a healthy lifestyle. Someone can still deliver the message of healthy lifestyle and be overweight.”.*
(Expert from Cambodia)

As the sixth step, we synthesized all data into the final guideline with the checklist. The final guideline is included in the Appendix A of this article. We expanded the final guideline to include background information as well as an introduction explaining the concept of ‘Positive Health’ and the importance of adjustment. Moreover, we added, as a recommendation, the process of cultural or contextual adaptation of CHBIs in three phases: the preparation phase, the assessment phase, and the adoption phase. These phases give guidance regarding the adaptation of materials and the implementation of CBHIs. Because the implementation of an intervention consists of more contextual aspects, such as the role of a trainer, compared to the adaptation of merely written intervention materials, the guideline includes two checklists: one for the materials and one for the implementation. The checklist for the implementation of an intervention is shown in Table 2. It includes 34 contextual and cultural aspects belonging to the topics: general aspects, the six dimensions of ‘Positive Health’, and the role of the implementer. Moreover, in the last step, we reflected upon the contextual aspects, using the conceptual framework; this showed that health perception is mostly influenced by aspects within the cultural and demographic context.

## 4. Discussion

We developed a guideline for the contextual adaption of CBHIs. We performed this in two stages, using seven steps derived from the AGREE-II tool. In the first stage, we used the conceptual framework of ‘Positive Health’ (based on determining contexts and health perception) to form the basis of a checklist for contextual adaptation. After a narrative literature review and informal meetings, we drafted our checklist. In the second stage, we developed a guideline (including the checklist), using a participatory action research approach involving pilot testing and multiple forms of co-creation with different stakeholders. 

This guideline is valuable for adapting existing or newly developed CBHIs to culture and context. To our knowledge, it is the first practical evidence-based guideline with a checklist to address the contextual adaptation of CBHIs with concrete examples and issues. From the literature, we know that a good fit between interventions and context requires careful adaptation [17]. The aim in developing the checklist was, thus, to create a tool to adapt CBHIs to context, which is necessary when implementing an existing CBHI in another area, country, or culture. According to Schloemer and Schröder-Bäck (2018), the transferability of existing interventions is complex, and a good fit between an intervention and the context is greatly affected by similarities and differences in the original and new contexts [35]. Transfer of interventions to other contexts has often been ineffective because such contextual aspects were ignored [17]. 

CBHIs are commonly developed using validated and evidence-based health directives or models from major international organizations, such as the World Health Organization or the Global Alliance for Chronic Diseases. Based on these generally validated health directives or models, the checklist and guideline developed as a result of our research has the potential to adapt CBHIs to the local context.

This guideline reflects the needs and knowledge of the involved stakeholders. In all steps of its development, we used a participatory action research (PAR) approach to shape co-creation, engaging community stakeholders as equal partners. Involving stakeholders is crucial in the process of contextual adaptation of interventions, as well as in the development of the final version of the checklist and guideline. This finding corresponds with other research on the implementation of interventions in a new context [36,37]. Stakeholder involvement has two overarching benefits. First, collaboration with stakeholders provides an awareness of context [38]; the engagement of various stakeholders led to the identification of contextual aspects that would have been missed if the checklist was based only on a literature review. When adapting a CBHI to a particular community, local stakeholders can help to harmonize it with the dynamics and structures of the community, incorporating its contextual aspects. Second, collaboration with stakeholders creates a feeling of ownership within the community involved [37,39]. Local ownership is especially important for CBHIs because it can enable the co-funding and sustainability of interventions [40]; this is often a challenge due to a lack of beneficiaries for community-based projects. The focus of PAR research is to make action possible; this is achieved through stakeholders collecting and analyzing data, and then determining together what action should follow [41]. 

Another important conclusion derived from our use of the PAR approach was that clear instructions were needed for optimal application of the checklist. The stakeholders, with their diverse levels of expertise and involvement in CBHIs, shared converging views and ideas for the construction of such instructions. As a result, this guideline can be more widely applied to other CBHIs and in other countries. Furthermore, as indicated in the literature, a guideline with a checklist for contextual adaptation of CBHIs can lead to more effective interventions to promote health [1,4,5,6], but not only health. All stakeholders indicated that the usefulness of this guideline extends beyond health-related community-based interventions.

Compared to similar guidelines, such as the ADAPT guideline [17] and the report of Graig and colleagues [18], we conclude that our guideline is a valuable addition to guidelines for the contextual and cultural adaptation of interventions. Our guideline is a practical and concrete guideline that can be assessed by different levels of involved people, e.g., intervention developers, implementers, observers, and/or trainers. Moreover, the guideline provides insight into the need for change in the already existing interventions. Using the checklists can be carried out periodically to check if changes are needed due to changing the context and/or culture. Compared to other guidelines, this can be carried out with little effort and time, making it a concrete and practical tool for the improvement as well as development of an intervention. 

We can conclude that our conceptual framework provides insight on how CBHIs can be adapted to the health perspectives of CBHI members. The concept of ‘Positive Health’ and its determining contexts guided the direction of our literature search and determined what we included in the checklist. Because ‘Positive Health’ is a broad concept: “the ability to adapt and self-manage” [14], based on a holistic view of aspects of life such as participation and daily functioning, we expect our guideline to be more generally applicable in community-based interventions, focused not only on health, but also on issues such as reducing natural disasters, preventing bullying among youth, and addressing gender violence.

### 4.1. Strengths and Limitations

A strength of this study is the use of various qualitative research methods to develop the guideline with the checklist. The use of multiple qualitative methods led to methodological triangulation, which provided increased validity and enhanced the understanding of contextual aspects [42]. The use of participatory action research led to generating collaborative knowledge, contributing to the scientific and practical base of the guideline. Finally, the process of co-creation is a further strength of this research: involving various stakeholders deepened and enriched the development of the guideline. 

A limitation of this study is that the guideline has been pilot tested only in Asian countries, and most of the involved stakeholders were from Asia. This may lessen its applicability to other continents. However, a number of involved stakeholders and an expert from continents other than Asia affirmed that the checklist and guideline were promising for application in their own country or continent. A final limitation is that the same stakeholders were involved in both the development and pilot testing of the guideline; this could have led to positive bias in the assessment of the pilot findings [43]. 

### 4.2. Implications

This guideline can be used for contextual adaptation of community-based interventions in Southeast Asia. As both Southeast Asian and non-Southeast Asian stakeholders have reviewed the guideline, we expect it to be potentially applicable in regions other than Southeast Asia. However, to confirm its applicability in other regions, the guideline should be further tested on interventions in other settings. 

The scientific conceptual framework, used as a basis for their development, imply that the checklist and guideline are likely to promote health more effectively when adapted optimally to context. The guideline was developed in a rigorous way, including multiple research methods and a diverse range of stakeholders. However, additional research is needed to assess their effectiveness and validity in routine practice.

This guideline has the potential to be a tool for more general contextual adaptation of interventions. Future research and practice should focus on its application beyond community-based health interventions. 

## 5. Conclusions

We have developed a guideline with a checklist for the contextual adaptation of CBHIs. To our knowledge, this is the first guideline to provide a practical and scientific base for contextual adaptation of both newly developed and already implemented CBHIs. Moreover, its guidance is based on general international health directives and models. 

## Figures and Tables

**Figure 1 ijerph-19-05790-f001:**
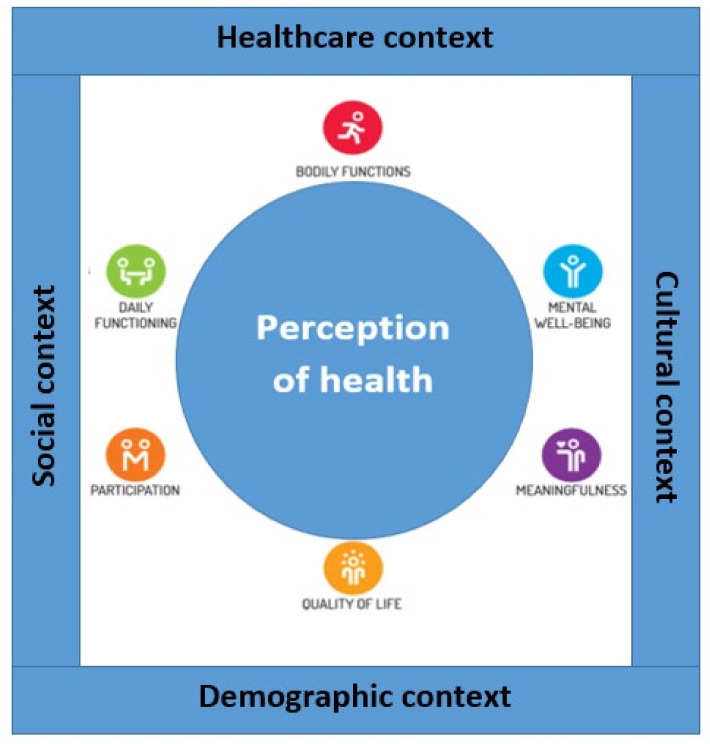
Conceptual framework: the six dimensions of health perception and its determining contexts. Based on Huber’s ‘Concept of Positive Health’ [14,15].

**Figure 2 ijerph-19-05790-f002:**
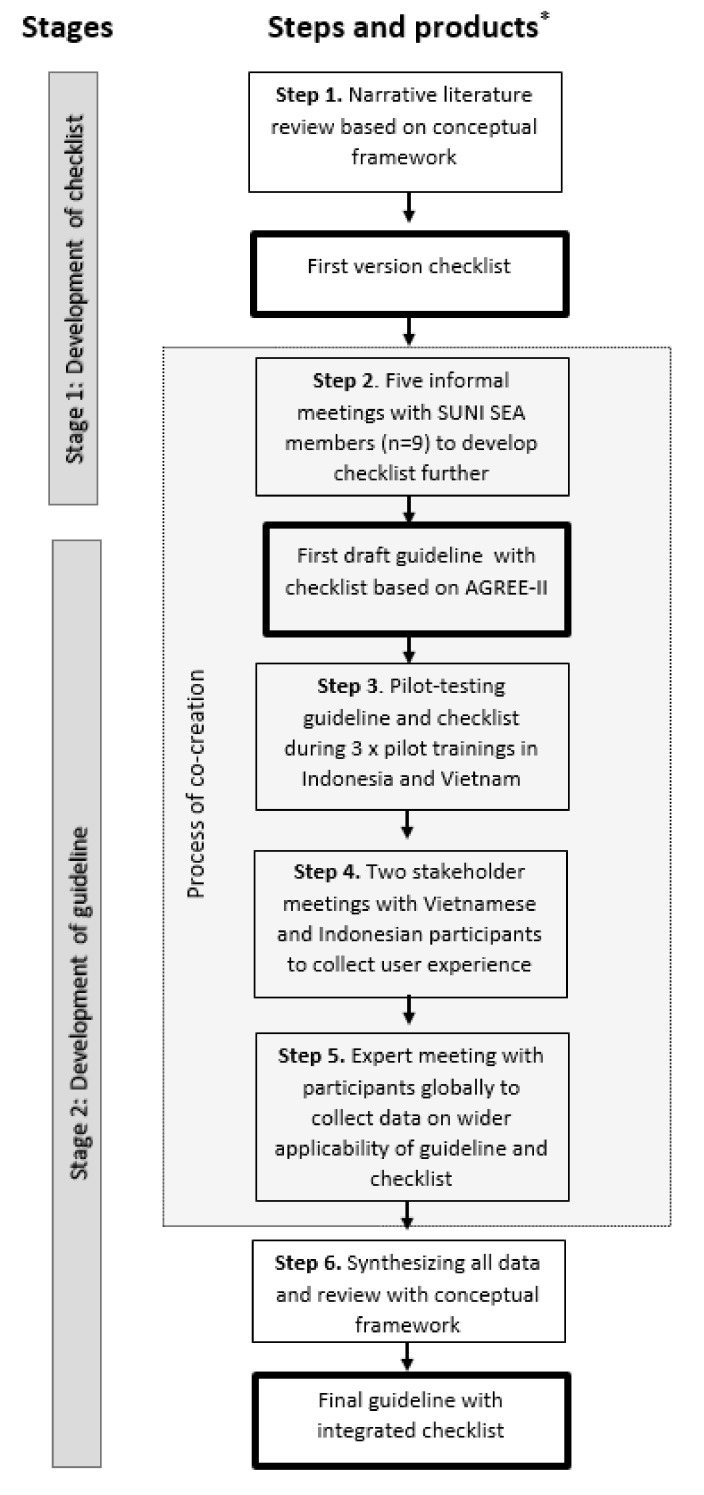
Design of the study structured in stages, steps, and products. * Boxes with bold outline denote products created in the process.

**Table 2 ijerph-19-05790-t002:** Checklist for cultural and contextual adaptation of community-based health interventions (CBHIs).

Topic	Contextual/Cultural Aspects	Yes	No
1. General	1a. Gender differences		
	1b. Ability to read/write		
	1c. Age friendly methods, addressing differences between generations; if end-users were adults, adult learning methods were applied		
	1d. Digital inclusion/exclusion		
2. Bodily functions	2a. Perception of own body		
	2b. Physical fitness (cultural and individual exercise options) and/or somatic complaints		
	2c. Coping with stress and stigmatization of illnesses		
3. Mental well-being	3a. Perceptions regarding health: individual differences		
	3b. Local health traditions		
	3c. Cultural influences in diet		
	3d. Cultural influences on healthy living		
	3e. Myths and facts regarding health promotion		
	3f. Stigmatization of mental health, main issues		
	3g. Psychological stress, sources		
	3h. Feeling supported: role of peers, working together on health		
	3i. Feeling of belonging: social cohesion, part of community		
	3j. Availability of/barriers to informal resources: relatives/friends		
	3k. Access to resources: Barriers to access healthcare and medicines		
	3l. Barriers to access health information		
4. Meaningfulness	4a. Religious and spiritual beliefs		
5. Participation	5a. Family structure: role of elders, in-laws and siblings		
	5b. Being able to participate, and having a role in usual community activities		
	5c. Being able to participate and having a role in usual family activities (earning money, cooking and cleaning)		
6. Daily functioning	6a. Availability of/barriers to healthy food		
	6b. Current/past working life		
7. Quality of life	7a. Social network, role of social structures in health, e.g., governmental and non-governmental organisations		
8. Role of implementer	8a. Does implementer represent or have knowledge of healthy lifestyle?		
	8b. Is implementer a role model for the target group?		
	8c. Is implementer culturally and linguistically matched to target group?		
	8d. Are participants treated equally and inclusively by implementer?		
	8e. No stigma or discrimination by implementer? Inclusiveness, stimulation of participants to come with solutions for local issues?		
	8f. Does implementer take into account cultural diversity of participants?		
	8g. Does implementer take into account different levels of participant knowledge?		
	8h. Does intervention enhance self-efficacy of participants?		
9. Lessons learned or other remarks:			

## Data Availability

All raw data are available upon request.

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
