# Peer review of "A Guideline for Contextual Adaptation of Community-Based Health Interventions"

_ijerph, 2022, doi:10.3390/ijerph19105790_

Round 1

Reviewer 1 Report

This is an interesting paper, describing a well thought-through process

I suggest minor revision.

No major concerns.

Some issues that would probably benefit from being addressed:

  1. Abstract: a narrative review was done (line 27). A review on what? (also true for line 105)
  2. Line 38 (keywords) maybe add CBHI
  3. Line 83 (and others). It is not clear to me what the conceptual framework is about. It is a conceptual framework about what?
  4. Line 43, ref 1: I am not sure this is a good reference. Ref 1 seems to refer to High-income country (note that your sentence – line 42 – starts with “Southeast Asia”). And it was more than 20 years ago (year 2001). Not e
  5. Line 43: you wrote: “CBHIs have been shown to promote health more effectively 24 when tailored to sociocultural aspects of health and well-being, like religion, traditions, and values.” I think this is note correct. You add references 2 to 4. At a maximum you can write: "there is some indication that CBHIs that are tailored to sociocultural aspects may promote health more effectively than when these aspects are not given attention to. Note that for example in terms of religion, references 2 (a systematic review!) and 3 do not mention it at all. Reference 4 mentions religion but does not present evidence. Based on this: please check (I did not check all) if you accurately used references and did not "exaggerate" existing evidence. 
  6. Line 86: You wrote: “We derived the steps per stage using”. Which steps? (it is not clear in this passage of your manuscript).
  7. Line 97 states that in total there were 29 stakeholders. Does the “in total” mean for ALL of the project, i.e. Stage 1 and Stage 2?
  8. Lines 171 to 172 and Appendix A. In appendix A you use “Aspects”. Maybe better to use “Dimensions”.
  9. I do not understand line 178 (what is the meaning of “contextual aspects for ‘daily functioning’ include current or past working life and availability of/barriers to healthy food.”? Is a word missing? What is working life? Is it “in current and past working activities”? (Maybe best to simplify it, e.g. “contextual aspects for ‘daily functioning’ include current or past activities during work”)
  10. Figure 2: I think it would be good to start the capture with “Conceptual framework: the six dimensions….”
  11. Line 317 is not clear to me. (“were involved in both the development pilot-testing of the guideline”)
  12. A minor detail: in some of the references, there are two or three spaces between words (rather than just one space). E.g . Line 373

Reviewer 2 Report

This is a very interesting and important study. Researchers in different settings have developed various interventions and compared with developing a new intervention, it would be a cost effective way to adapt the existing interventions to other settings. Cultural adaptation is a very crucial element when we transfer an existing intervention to another setting that is different from the original setting the intervention was developed. In this study, a guideline and checklist were developed for the contextual adaptation of community-based health intervention based on the concept of ‘Positive Health’ through a narrative literature review, stakeholder meetings, expert meetings and pilot tests. This work is novel and has important implications for behaviour change for better prevention and management of NCDs. Some comments are listed below for the authors' reference:

  1. It would be more informative if the abstract could contain more of a description of the methodology, and a description of the results needs to be added; please describe the content of the guidelines checklist: e.g., number of topics, number of entries in terms of contextual/cultural aspects, etc.
  2. The inclusion and exclusion criteria for the narrative literature review are not given in this manuscript. The critical appraisal process of the included studies were not described either. Are there any limitations on the research design of the included studies and what is the retrieval time of database search? Is there a language requirement? Does the number of included articles mean that only 33 relevant articles were searched or is this number a filtered process? Please describe more about the review process;
  3. The results of the literature review are the basis for developing the first version of the checklist for this study, and I recommend that the authors list the summary of articles identified through the search strategyï¼›
  4. Some descriptions of the "Design" and "Sample and procedure" sections are duplicated, please refine and integrate them;
  5. From the current description of this manuscript, it is only possible to understand that stakeholder meetings took place, but the content of the meetings are not clearly presented, so please provide more details of the stakeholder meetings;
  6. In Materials and Methods 2.4-, it is confusing whether the stakeholders in step 5 are the participants in the pilot test or who were working on the development, research and implementation of CBHIs?
  7. Stakeholders and experts participation and involvement in the study are very importance base for guideline development; however, the background information of the stakeholders/experts are very limited and the number of the stakeholders/experts is relatively small. Please give more information about the stakeholders/experts.
  8. Please unify the reference format. 

Reviewer 3 Report

This paper drafted a contextual adaptation of CBHIs by 24 stakeholders in five informal meetings, two stakeholder meetings, and an expert review meeting. However, it is needed an effective test for the guideline to examine its appropriateness. Many suggestions as the following:

  1. the introduction lacks of the description   to the current guidelines and limits to the CBHIs, it should add literature review.
  2. the research design in figure 1 seems as a study flow chart rather than a theoretical framework (research design concept). It is difficult to understand the author's theoretical concept to the guideline.
  3.  the practice of the guideline application is unclear to examine the effectiveness and appropriateness of the guideline.
  4.  the structure of the stakeholders and experts should describe and exam their appropriateness to the data collection.

Reviewer 4 Report

Thank you for submitting this paper. After reading your paper, I found that this paper must be improved before it can be published. 

Fisrt of all, I think the introduction is too short, making readers (particular international readers) unable to understand the importance of the topic.

The authors failed to discuss how the form (the items/questions) were formed. I expected this would be discussed in the introduction/literature review. Such checklist/guideline/theory (whatever you call) should be developed based on previous research (e.g. Positive Health that you mentioned).

Results: Comments from stakeholders should be discussed in detail. Readers expect findings and analyses here, rather than steps of froming the results. 

The discussion and conculsion are too short and they elebrate insufficent information. The discussion and conclusion are not supported by the results anyways as the results did not present well.

This journal has no word limitation so the authors should provide information as details (and in depth) as possible. 

I hope the authors can revise this paper thorughout and resubmit it. 

All the best. 

Round 2

Reviewer 2 Report

The authors have improved the manuscript considerably. All responses are acceptable to me. Please check the full text carefully to avoid confusion caused by inconsistencies in the presentation.

1. The number of stakeholders in the abstract (24) does not match that in Figure 2 and in the text (29). The sum of interviewees from five informal meetings, two stakeholder meetings and one expert meeting should be 29.

2. Line 263 "In the second stage we developed a guideline." would be more appropriately placed after the second paragraph of 3.1 so that it corresponds to the methodology section.

Author Response

Dear Dr. Annemarie Wagemakers and dr. Lea den Broeder Editors of the Special issue  Community Participation in Health Promotion: Challenges and Successes

Thank you for again inviting us to revise our manuscript (ijerph-1671174), entitled “A guideline for contextual adaptation of community-based health interventions” potentially acceptable for publication conditional on minor revisions. We have carefully considered the comments and revised the manuscript accordingly. Please find enclosed a detailed response, preceded by [Response] to all comments.

We thank the reviewers again for their clarifying comments and suggestions. We hope that the revised version can contribute to the contents of International Journal of Environmental Research and Public Health.

With kind regards,

Also on behalf of the co-authors

Zinzi E. Pardoel

Reviewer 1

You replied to the comments.

[Response] Thank you very much for reviewing the revised manuscript and response letter. We are pleased to hear that we have addressed your concerns and recommendations to your satisfaction.

Reviewer 2

Comments and Suggestions for Authors

The authors have improved the manuscript considerably. All responses are acceptable to me. Please check the full text carefully to avoid confusion caused by inconsistencies in the presentation.

[Response] Thank you very much for reviewing the revised manuscript and response letter. We are pleased to hear that you find the responses acceptable. We have checked the full text carefully and revised inconsistencies in the presentation. For example, in the manuscript we inconsistently used apostrophes for the concept of Positive Health. We have revised this into using apostrophes consistently in the manuscript, for example in lines 152 and 199.

  1. The number of stakeholders in the abstract (24) does not match that in Figure 2 and in the text (29). The sum of interviewees from five informal meetings, two stakeholder meetings and one expert meeting should be 29.

[Response 1] Apologies for this inconsistency. We have revised 24 to 29 in line 39.

  1. Line 263 "In the second stage we developed a guideline." would be more appropriately placed after the second paragraph of 3.1 so that it corresponds to the methodology section.

[Response 2] Thank you for this suggestion. We have replaced line 263 to line 258. We agree that this corresponds to the methodology section.

Reviewer 3

Comments and Suggestions for Authors

The author have revised the manuscript completely and can be accepted for publication without further revision.

[Response] Thank you very much for reviewing the revised manuscript and response letter. We are pleased to hear that we have addressed your concerns to your satisfaction and have revised the manuscript completely.

Reviewer 4

Comments and Suggestions for Authors

The authors have addressed all my questions and revised the manuscript accordingly. I am happy to accept this paper.

[Response] Thank you very much for reviewing the revised manuscript and response letter. We are pleased to hear that we have addressed all your questions and revised the manuscript to your satisfaction.

Reviewer 3 Report

The author have revised the manuscript completely and can be accepted for publication without further revision.

Author Response

(The authors gave the same response as above.)

Reviewer 4 Report

The authors have addressed all my questions and revised the manuscript accordingly. I am happy to accept this paper. 

Author Response

(The authors gave the same response as above.)
